

# Changes in anti-nutrient, phytochemical, and micronutrient contents of different processed rubber (*Hevea brasiliensis*) seed meals

Chidinma M. Agbai[1,*], Ijeoma A. Olawuni[1,*], Chigozie E. Ofoedu[1,2], Chidi J. Ibeabuchi[1], Charles Odilichukwu R. Okpala[3], Ivan Shorstkii[4] and Małgorzata Korzeniowska[3]

[1] Department of Food Science and Technology, School of Engineering and Engineering Technology, Federal University of Technology, Owerri, Owerri, Imo, Nigeria
[2] School of Food Science and Engineering, South China University of Technology, Guangzhou, Guangdong, China
[3] Department of Functional Food Products Development, Faculty of Biotechnology and Food Science, Wrocław University of Environmental and Life Sciences, Wroclaw, Poland
[4] Department of Technological Equipment and Life-Support Systems, Kuban State Technological University, Krasnodar, Russian Federation
* These authors contributed equally to this work.

Corresponding authors
Chigozie E. Ofoedu, chigozie.ofoedu@futo.edu.ng
Charles Odilichukwu R. Okpala, charlesokpala@gmail.com

## ABSTRACT

Rubber (*Hevea brasiliensis*) is a perennial plant crop grown in many parts of Africa, South East Asia, and South America, especially within the hot and humid climatic regions. Rubber seed, either as feed or food, is a useful raw material to produce edible oil and protein. Despite the huge quantity of rubber seeds produced in Nigeria and its potential as a protein source, rubber seeds still appear neglected and under-utilised as feed/food given its perception as inedible and toxic due to the high concentration of cyanogenic glycoside. Therefore, the quest for effective processing technique(s) that would enhance its food use application is very fitting. This current study was directed to determine the changes in anti-nutrient, phytochemical, and micronutrient contents of different processed rubber seed meals. Specifically, the rubber seeds underwent processing, which employed boiling and the combined action of boiling and fermentation methods that brought about three seed meal flour groups, i.e., raw (RRSM), boiled (BRSM), and fermented (FRSM) seed meals. These were subsequently analysed for anti-nutrient/phytochemical (oxalate, phytate, tannin, phenols, saponin, hydrogen cyanide (HCN), alkaloids, flavonoids, and trypsin inhibitors), and micronutrient (which involved minerals (magnesium, phosphorus, calcium, iron, zinc, potassium, sodium, manganese, lead, and selenium) and vitamin (vitamin B1, B2, B3, C, E, and beta carotene)) contents. The results showed that the processing methods used to achieve the RRSM, BRSM, and FRSM, reduced the anti-nutrients (phytate, tannin, and oxalate) below the acceptable limits, and the HCN below the toxic levels. Importantly, the processing methods herein have not yet succeeded in removing HCN in the (processed) rubber seed meals, but can be seen to be heading toward the right direction. The FRSM obtained significantly lower ($p < 0.05$) anti-nutrient/phytochemical, but significantly higher ($p < 0.05$) mineral contents, compared with the other groups (RRSM and BRSM), except for flavonoids that obtained a 30% increase over the BRSM. Some mineral and

vitamin contents could be lost in the BRSM compared to the others (RRSM and FRSM) in this study. Additionally, the FRSM obtained higher vitamin contents, after those of RRSM. Overall, the combined action of boiling and fermentation should be recommended for the proper utilisation of rubber seed as food/feed.

# INTRODUCTION

Conventional agriculture has made cereal, legume, pulse, and meat food-protein products possible for human consumption. Unfortunately, today's sky-rocketing demand for protein-rich meals has put the conventional production of the above-mentioned (agrofood products) at risk, which has arisen owed to the inability of conventional production to cope, given by the increases in today's global (animal/human) population. Importantly, over 8.9% of the global human population which suffers from hunger and malnutrition especially protein deficiency, has been mainly attributable to the high cost of meat and protein-rich foods (*Medek, Schwartz & Myers, 2017*; *Drago, 2017*; *Redondo et al., 2019*; *FAO, IFAD, UNICEF, WFP & WHO, 2020*). It is equally true that such increases in the global (animal/human) population bring about the concept of food insecurity. A sustainable nutrient-dense agrofood source is therefore achieved through increased research to develop healthy, quality, and safe (agrofood) products, particularly those that can be made from more eco-friendly processes, which would essentially be adapted to the variants of consumers' needs (*Bigliardi & Galanakis, 2020*). A potential solution can be to identify a nutrient-dense agrofood source that is cheap, available, and less competed by animal, man, and industry (*Akinmutimi, 2004*).

Rubber (*Hevea brasiliensis*) is a perennial plant crop grown in many parts of Africa, South East Asia, South America, and regions with hot and humid climates. According to the Nigerian Rubber Research Institute, about 28,000 tonnes of rubber seeds are estimated from 354,000 ha of rubber plantations in Nigeria (*FAOSTAT, 2019*; *Maliki & Ifijen, 2020*). Rubber tree is commercially grown in Nigeria for latex production, while the abundant seeds are under-exploited and wasting. Rubber seed meal has been reported to have considerable amounts of absorbable nutrients than many conventional seed meals (*Atasie, Akinhanmi & Ojiodu, 2009*; *Aguihe et al., 2017*; *Udo, Ekpo & Ahamefule, 2018*) as high amount of protein content (19.40–30.68%), crude fat (42.50–54.17%) and carbohydrates (11.58–29%) had been reported from earlier studies (*Onwurah et al., 2010*; *Sharma, Saha & Saha, 2014*; *Hossain et al., 2015*; *Suprayudi et al., 2015*; *Lalabe, Olusiyi & Afolabi, 2017*; *Agbai et al., 2020*). This is also similar to the composition of some conventional seed meals such as soybean meal and cottonseed meal as reported by *Tang et al. (2012)* and *Ma et al. (2019)*. As a result of the enormous potential of rubber seed as an oil and protein-rich source, multiple studies have reported extraction of oil from rubber seed (*Morshed et al., 2011*; *Singh, Yusup & Wai, 2016*; *Widyarani et al., 2017*) for

biodiesel production and other purposes, and derivation of protein isolate from rubber seed meal (particularly after oil separation) (*Fawole et al., 2016b*; *Widyarani et al., 2017*; *Fawole et al., 2016a*) for protein valorization, feed, and food uses. Therefore, this nutrient-dense (rubber seed meal) product can influence food composition by proportion incorporation during product formulation/development. Besides processed rubber seed being consumed in some regions of Indonesia as a staple diet (*Lukman, Nurul & Connie, 2018*) and in Malaysia as a daily dish (*Asam rong*) (*Siti et al., 2013*), there are continued hindrances to its utilization as food in many parts of the world, largely due to the toxicity of the rubber seed, given by the high concentration of anti-nutrients (phytochemicals) and cyanogenic glycoside (*Basher & Jumat, 2010*). Despite the interference offered by anti-nutrients (phytochemical compounds) in nutrient absorption, which act to reduce the nutrient intake, digestion, and utilization (*Popova & Mihaylova, 2019*), the exposure to cyanide from intentional or unintentional consumption of food with a high dose of cyanogenic glycoside could lead to acute intoxication characterized by growth retardation and neurological symptoms resulting from tissue damage in the central nervous system (*Bolarinwa et al., 2016*).

There is ample evidence in relevant literature that rubber seed products have been of interest to many researchers. Examples of areas of focus can include nutrient value of rubber seed flour (*Onwurah et al., 2010*), processing effects on nutrient composition of rubber seed meal (*Udo, Ekpo & Ahamefule, 2018*), and basic properties of crude rubber seed oil (*Yusup & Khan, 2010*). Indeed, the rubber seed remains posited with a high content of cyanogenic glycosides (*Eka, Tajul Aris & Wan Nadiah, 2010*; *Oluodo, Huda & Komilus, 2018*) like those associated with cassava tubers. Through processing and treatment methods, cyanogenic glycoside can either be eliminated or reduced successfully, and examples applied in the context of rubber seed include heat treatments (toasting, boiling, roasting), storage (2–4 months) at ambient temperature (*Eka, Tajul Aris & Wan Nadiah, 2010*; *Udo, Ekpo & Ahamefule, 2018*; *Oluodo, Huda & Komilus, 2018*; *Farr et al., 2019*), enzymatic reaction, absorbent (*Oluodo, Huda & Komilus, 2018*), and fermentation (*Syhruddin, Herawaty & Ningrat, 2014*; *Oluodo, Huda & Komilus, 2018*; *Farr et al., 2019*). There is a high chance that, through such (above-mentioned) processing treatment methods, different grades of rubber seed meals can be actualised. However, despite these various processing methods previously used to treat rubber seed, there is still a paucity of relevant literature, particularly on how the combination of treatment methods, for instance, heat treatments and fermentation, could influence the chemical components of rubber seed meal. It is possible that, through such combined treatment methods, both processors and product developers on (seed meal) selection could achieve promising quality output, which could enhance and strengthen the edibility of processed rubber seed meals. To supplement existing information, therefore, this current study investigated the changes in anti-nutrient, phytochemical, and micronutrient contents of different processed rubber seed meals. It is expected that the combined action of boiling/heat treatment and fermentation process would help elevate the edibility and resultant quality of the rubber seed product.
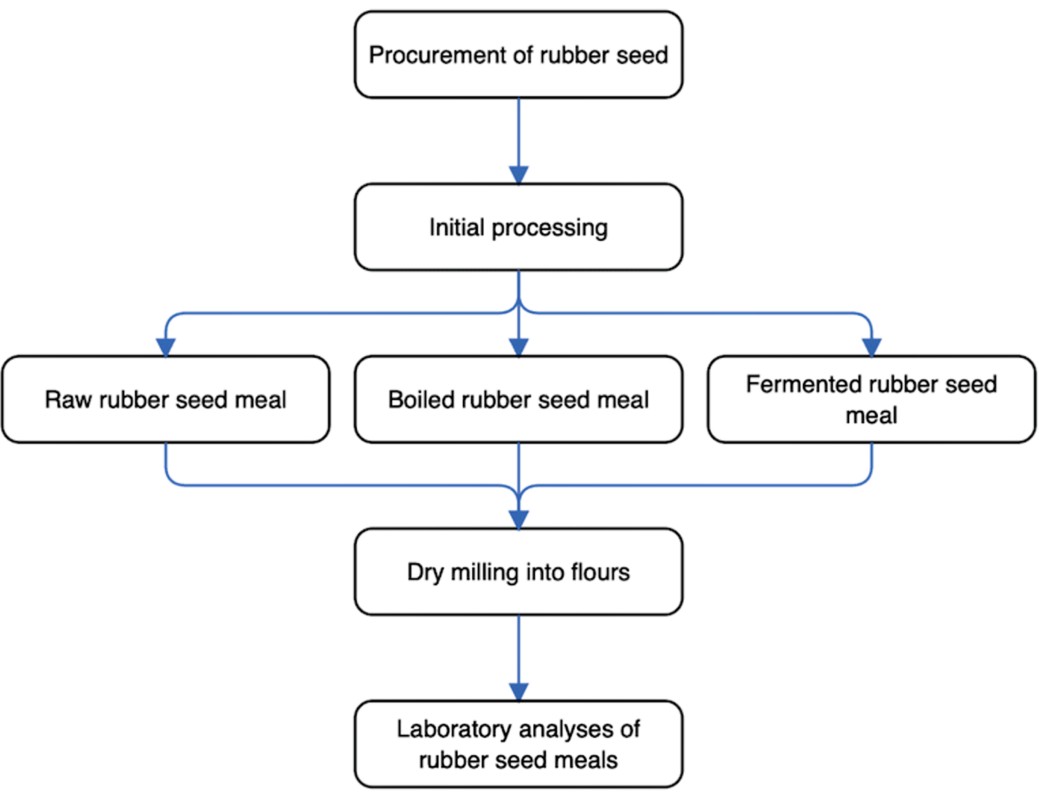

**Figure 1 Schematic diagram of experimental program.** The resultant seed meal were categorized as raw rubber seed meal (RRSM), boiled rubber seed meal (BRSM) and fermented rubber seed meal (FRSM).

## MATERIALS AND METHODS

### Schematic overview of the experimental program

The experimental program schematically depicting the key steps, from the procurement of rubber seed samples to laboratory analyses of (rubber seed) meals, is shown in Fig. 1. In particular, this current research was designed to determine the changes in anti-nutrient, phytochemical, and micronutrient contents of different processed rubber seed meals. Specifically, the rubber seeds underwent processing, which brought about three seed meal flour groups: (a) raw rubber seed meal (RRSM); (b) boiled rubber seed meal (BRSM); (c) fermented rubber seed meal (FRSM). Subsequently, the emergent rubber seed meal flours were analyzed for anti-nutrients, micronutrients, and phytochemicals content.

### Chemicals and reagents

All chemicals and reagents (i.e., Acetic acid ($CH_3COOH$), Alcoholic potassium hydroxide ($C_2H_7kO_2$), Ammonia solution ($NH_4OH$), Aqueous ethanol solution ($C_2H_5OH$), Copper (II) sulphate ($CuSO_4$), Diethyl ether ($C_2H_5OH$), Erichrome black T ($C_{20}H_{12}N_3O_7SNa$), Ethanolic sodium hydroxide ($C_2H_7NaO_2$), Ethyl acetate ($CH_3COOC_2H_5$), Ethylenediaminetetra acetic acid (EDTA) ($C_{10}H_{14}N_2Na_2O_8$), Ferric chloride ($FeCl_3$), Folin ciocalteau reagent (Phosphomolybdate and phosphotungstate mixture), Folin Denis

reagent (Phosphomolybdate and phosphotungstate mixture), Gallic acid solution $(C_6H_2(OH)_3COOH)$, Hydrochloric acid (HCl), Hydrogen peroxide $(H_2O_2)$, Hydrogen sulphate $(H_2SO_4)$, Hydroxylamine hydrochloride $(NH_2OH.HCl)$, N- hexane $(C_6H_{14})$, N-butanol $(C_4H_9OH)$, N-$\alpha$-benzoyl-DLarginine-P-nitroanilide (BAPA), Phosphorus colour reagent (vanadomobydate solution), Picrate, Potassium cyanide (KCN), Potassium dichromate $(K_2Cr_2O_7)$, Potassium iodide solution (KI), Potassium permanganate solution $(KMnO_4)$, Potassium chloride (KCl), Sodium carbonate solution $(Na_2CO_3)$, Sodium chloride (NaCl), Sodium cyanide (NaCN), Sodium hydroxide solution (NaOH), Sodium potassium ferrocyanide $(Na_4Fe(CN)_6)$, Sodium sulphate $(Na_2SO_4)$, Solochrome dark blue indicator $(C_{20}H_{13}N_2O_5SNa)$, Standard phosphorus solution $(K_2HPO_4)$, $\alpha$-$\alpha$-dipyridyl., and Standard tannin acid solution $(C_{76}H_{56}O_{46})$ used in this current study were of analytical grade standard.

## Processing of rubber seed meal samples

Specifically, mature rubber seeds were collected from Nigerian Rubber Institute, Akwete, Abia state, Nigeria. Processing of rubber seed samples (1 kg) which involved sorting, dehulling, washing of seeds, boiling, and fermentation of rubber seeds were implemented at the Department of Food Science and Technology, Federal University of Technology—Owerri, Nigeria. The schematic diagram for the production process to generate the rubber seed meals is presented in Fig. 2. As shown, the raw rubber seed meal (RRSM) was produced by drying rubber seeds in an oven (Model M 30 C, S/N 92B060; Genlab, Cheshire, England) at 60 °C followed by milling using a blender (Model BL330; KenWood Blender, UK) into a fine flour. The boiled rubber seed meal (BRSM) was produced by boiling the rubber seeds for 2 h, followed by oven drying at 60 °C for 24 h and dry milling into a fine flour. The fermented rubber seed meal (FRSM) was produced by wrapping 2 h of boiled rubber seeds in already steam blanched plantain leaves. Subsequently, the wrapped sample was placed in a basket to ferment for 3 days at a temperature between 30–35 °C, followed by oven drying at 60 °C for 24 h and dry milling into a fine flour. Flours obtained from raw and processed (boiled and fermented) rubber seed meals were tightly sealed in glass containers until further analysis.

## Anti-nutrient, micronutrients, and phytochemical measurements of different rubber seed meals

Determination of some anti-nutrient, micronutrient, and phytochemical contents of rubber seed meals was conducted at the Department of Food Science and Technology, Federal University of Technology, Owerri, Nigeria, as well as Food Science laboratory at Rivers State University of Science and Technology, Port Harcourt, Nigeria.
For emphasis, the anti-nutrients/phytochemicals determined include; oxalate, phytate, tannin, phenols, saponin, hydrogen cyanide (HCN), alkaloids, flavonoids and trypsin inhibitors (TI), while the micronutrients include minerals and vitamins. Specifically, the minerals determined were magnesium (Mg), phosphorus (P), calcium (Ca),

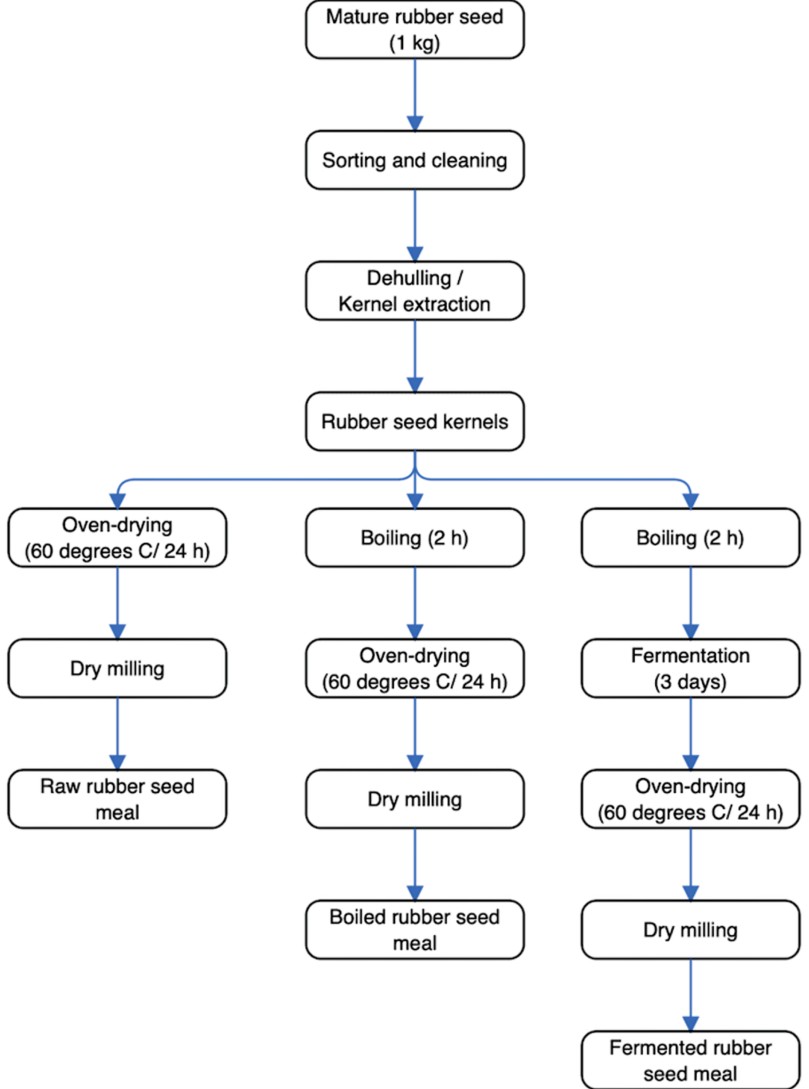

**Figure 2 The schematic diagram for the production process to generate the rubber seed meals.** The processes that the rubber seed underwent, to bring about raw, boiled and fermented rubber seed meal can be seen.

iron (Fe), zinc (Zn), potassium (K), sodium (Na), manganese (Mn), lead (Pb), and selenium (Se). Specifically, also, the vitamins determined include thiamine (vitamin B1), riboflavin (vitamin B2), niacin (vitamin B3), beta carotene, ascorbic acid (vitamin C), and tocopherol (vitamin E). Samples used in the analysis were chosen by simple random (aliquot) sampling from the population of the rubber seed meals (RRSM, BRSM, or FRSM). Independently and by analysing aliquot samples, duplicate determinations were carried out from the sample population (RRSM, BRSM, or FRSM) of rubber seed meals.

## Determination of anti-nutrient/phytochemical contents of rubber seed meals

### a) Oxalate and phenol of rubber seed meal

The oxalate content of rubber seed meal was determined using the titration method described by *Agbaire (2011)*, which involved mixing test sample (1 g) and 3 M $H_2SO_4$ on a magnetic stirrer for 1 h, followed by filtration, then the filtrate titrated against 0.05 M $KMnO_4$ solution until a faint pink colour persisted for at least 30 s. The oxalate content was then calculated by taking 1ml of 0.05 M KMnO4 as equivalent to 0.63 mg oxalate.

The phenol content of rubber seed meal was determined using the colorimetric method described by *Henriquez et al. (2010)*, which involved the reduction of Folin-Ciocalteau reagent by Phenolic compounds with a concomitant formation of a blue complex, followed by taking absorbance reading using UV-VIS Spectrophotometer Model 6305 (Bibby Scientific Ltd, UK) at a wavelength of 765 nm. The measurement was compared to a standard curve prepared with Gallic Acid solution. The total phenolic content (TPC) was expressed as milligrams of Gallic Acid Equivalents (GAE) per gram of fresh weight.

$$\text{Total phenolic content} = \frac{C \times V}{W} \qquad (1)$$

Where C = concentration of gallic acid calculated from the calibration curve (mg/ml)

    V = volume of extract (ml)

    W = weight of the sample (g)

### b) saponin, flavonoid, and alkaloid of rubber seed meal

Flavonoid and saponin content was determined using the method described by *Okwulehie et al. (2017)*, while alkaloid content was determined using the method of *Onwuka (2018)*. Whilst saponin employed double solvent extraction gravimetric method, alkaloid employed precipitation gravimetric method. Specifically, flavonoid, saponin, and alkaloid content expressed as mg/100g were calculated using the equation below;

$$\text{Flavonoid/Saponin/Alkaloid (mg/100 g)} = \frac{W2 - W1}{\text{Weight of sample}} \times 100 \qquad (2)$$

Where; $W_1$ = Weight of filter paper

    $W_2$ = Weight of filter paper plus flavonoid/saponin/alkaloid precipitate

### c) Tannin, HCN, phytate and trypsin inhibitor of rubber seed meal

The tannin, HCN, phytate, and trypsin inhibitor (TI) content of rubber seed meals were determined using UV-VIS Spectrophotometer Model 6305 (Bibby Scientific Ltd, UK) consistent with the method of *Onwuka (2018)*. Further, the absorbance of tannin, HCN, phytate, and trypsin inhibitor was read at a wavelength of 250 nm, 490 nm, 519 nm, and

410 nm respectively. The amount of measured tannin, HCN, and phytate from rubber seed meals expressed in mg/100g was calculated using the equation below;

$$\text{Tannin/Phytate/HCN (mg/100 g)} = \frac{Au \times C \times Vf \times 100}{W \times As \times Va} \qquad (3)$$

Where Au = absorbance of the test sample
  As = Absorbance of standard solution
  C = concentration of standard solution
  W = Weight of sample used
  Vf = Total volume of extract
  Va = Volume of extract
  On the other hand, one trypsin unit inhibited (TUI) is equal to an increase of 0.01 absorbance unit at 410nm. Therefore, trypsin inhibitor (TI) activity expressed as the number of trypsin units inhibited (TUI) per unit weight of the sample analysed was calculated using the equation below;

$$\text{Trypsin inhibitor (TUI/mg)} = \frac{Au \times 0.01 \times F}{As} \qquad (4)$$

Where Au = Absorbance of the test sample,
  As = Absorbance of standard (uninhibited) sample
  F = Experimental factors
Where experimental factor, F is expressed as

$$F = \frac{Vf}{Va \times W} \qquad (5)$$

Where Vf = Total volume of extract
  Va = Volume of extract analyzed
  W = Weight of sample analyzed.

## Determination of micronutrients of rubber seed meals
### a) Mineral contents of rubber seed meals
The mineral contents (Mg, P, Ca, Fe, Zn, Mn, Pb, and Se) of rubber seed meal samples were determined, which entailed initial digestion in HCl, and subsequently, the use of atomic absorption spectrophotometry (AAS) Model 200A (Buck Scientific Inc., Norwalk, CT, USA), consistent with the AOAC method (*AOAC, 2003*). This involved background correction using an inserted lamp, internal flow of inert gas during charring, and properly regulated drying as liquid injections were micro-pipetted into the furnace. Analytical conditions were set as per the design manufacturer. On the other hand, other mineral contents (Na and K) of rubber seed meal samples were determined by initial calibration of Flame Photometer (FP) Model PFP-7 (Buck Scientific Inc., Norwalk, CT, USA) using standard diluted element (Na & K) solution to generate a standard curve, consistent with the method of *AOAC (2003)*. Subsequently, the test sample was measured and each test element was extrapolated from the standard curve. When the analytical process ended, mineral content results were reported.

### b) Vitamin contents of rubber seed meals

The vitamin B1, B2, and B3 content of rubber seed meals were determined using UV-VIS Spectrophotometer Model 6305 (Bibby Scientific Ltd, Stone, UK) consistent with the method of *AOAC (2005)*. The absorbance of vitamin B1, B2, and B3 was read at a wavelength of 360, 510, and 470 nm respectively. The amount of measured vitamin from rubber seed meals expressed in mg/100 g was calculated using the equation below:

$$\text{Vitamin B1/B2/B3 (mg/100 g)} = \frac{100 \times Au \times C \times Vt}{W \times As \times Va} \tag{6}$$

Where W = weight of sample ash (g)

    Au = absorbance of the test sample

    As = absorbance of standard thiamine solution

    C = concentration of standard thiamine solution (mg/ml)

    Vt = total extract volume (ml)

    Va = volume of extract analysed (ml)

The β-carotene content of rubber seed meals was determined using the method described by *AOAC (2003)*, which involved test sample (5 g) homogenized in acetone solution, filtered (Whatman No. 1 filter paper, Merck KGaA, Darmstadt, Germany), and an aqueous solution containing β-carotene was extracted from the filtrate using petroleum spirit. The absorbance of the solution was read using UV-VIS Spectrophotometer Model 6305 (Bibby Scientific Ltd, Stone, UK) at a wavelength of 450 nm. The amount of β-carotene expressed in mg/100g was calculated using the equation below;

$$\text{Carotene content (mg/100 g)} = \frac{A \times Volume(ml) \times 1000}{Ac \times Sample\ weight(g)} \tag{7}$$

Where; A= Absorbance

    $10^3$ = dilution factor

    $A_c$ = extinction coefficient of 1% β-carotene solution at 450 nm

The vitamin c content of the rubber seed meal was determined according to the method described by *Okwu & Ndu (2006)*, which involved test sample (2 g) homogenized in EDTA solution, filtered, and subsequently passing the filtrate through packed cotton wool containing activated charcoal to remove the colour. Potassium iodide and starch (indicator) solution were added to the filtrate and the mixture titrated against 0.01 M $CuSO_4$ solution until an endpoint marked by black specks at the brink of the mixture is reached. The vitamin C content was given by the relationship that 1 ml of 0.01 M $CuSO_4$ equals 0.88 mg vitamin C and therefore, calculated thus;

$$\text{Vitamin C (mg/100 g)} = \frac{100 \times Vf \times 0.88T}{W \times Va} \tag{8}$$

Where W = weight of the sample (g)

    Vf = total volume of extract (ml)

    Va = volume of sample in extract (ml)

    T = titre value

The vitamin E content of rubber seed meal was determined using UV-VIS Spectrophotometer Model 6305 (Bibby Scientific Ltd, Stone, UK) consistent with the method of *Achikanu et al. (2013)*, which involved macerating test sample (1 g) in 20 ml n-hexane, followed by centrifuging (Yescom 800-1 Centrifugal machine, Model No: 35CEN001-800-09, China) at 1,500 rpm for 10 min. The centrifuged mixture was filtered, treated with 0.5N alcoholic potassium hydroxide and ethanol, followed by evaporating filtrate (2 ml) to dryness over a boiling water bath. Subsequently, the residue was mixed with ethanol, 0.2% ferric chloride, and 0.5% $\alpha$-$\alpha^1$-dipyridyl, then the absorbance was taken at 520 nm against the blank (distilled water prepared with the procedures as the sample). The concentration of vitamin E expressed in mg/100g was calculated below;

$$C_x = \frac{Ax \times Cs}{As} \tag{9}$$

Where; Ax = Absorbance reading of the sample

As = Absorbance reading of the standard

$C_s$ = Concentration of the standard

## Statistical analysis

The analysis of variance (ANOVA) assumptions considered homogeneity of variances and normality, which were based on Levene's, and Shapiro–Wilk tests respectively (*Ofoedu et al., 2020*). One-way analysis of variance (ANOVA) was applied to the data obtained from duplicate measurements of samples. Results of dependent variables were expressed as mean ± standard deviation (SD). Fisher's least significant difference (LSD) was used to resolve mean differences. For all analyses, the probability level of statistical significance was set at $p < 0.05$ (95% confidence interval). IBM SPSS software version 20 (IBM Corporation, New York, NY, USA) was used to do the analysis.

# RESULTS

## Variations in anti-nutrient/phytochemical contents

Variations in anti-nutrient and phytochemical contents of rubber seed meal sample are shown in Table 1. Across the RRSM, BRSM, and FRSM samples, the anti-nutrient and phytochemical (namely: phytate, oxalate, tannin, saponin, HCN, TI, alkaloids, phenols and flavonoids) differed significantly ($p < 0.05$). Largely, a significantly ($p < 0.05$) decreasing anti-nutrient and phytochemical composition trend seems obvious across the RRSM, BRSM, and FRSM samples. However, this appears not strictly so for the flavonoids, which in FRSM (52.50 mg QE/100 g) was significantly ($p < 0.05$) above that of BRSM (40.00 mg QE/100 g). Across the RRSM, BRSM, and FRSM samples, the anti-nutrient and phytochemical composition obtained diverse ranges, as shown in phytate (from 19.62 to 6.97 mg/100 g), oxalate (from 13.26 to 3.36 mg/100 g), tannin (from 8.98 to 0.92 mg/100 g), saponin (from 4.91 to 1.60 mg/100 g), HCN (from 12.41 to 1.97 mg/100 g), TI (from 8.43 to 0.97 TiU/mg), alkaloids (from 4.54 to 0.97 mg/100 g), phenols (from 2.77 to 0.50 mg GAE/ g), and flavonoids (from 60.00 to 40.00 mg QE/100 g) contents. Also

**Table 1 Variations in anti-nutrients and phytochemical composition of rubber seed meal sample.**

| Parameters | *RRSM | *BRSM | *FRSM | **Safety limit (mg/100g) | **Sources for the safety limits |
|---|---|---|---|---|---|
| Phytate (mg/100 g) | 19.62[a] ± 0.22 | 8.82[b] ± 0.10 | 6.97[c] ± 0.10 | <25 | *Abdoulaye, Brouk & Jie (2011)* |
| Oxalate (mg/100 g) | 13.26[a] ± 0.02 | 7.54[b] ± 0.09 | 3.36[c] ± 0.06 | <10 | *Mark (2020)* |
| Tannin (mg/100 g) | 8.98[a] ± 0.06 | 2.21[b] ± 0.06 | 0.92[c] ±0.06 | 12 | *Ndie & Okaka (2018)* |
| Saponin (mg/100 g) | 4.91[a] ± 0.01 | 2.39[b] ± 0.01 | 1.60[c] ± 0.04 | | |
| HCN (mg/100 g) | 12.41[a] ± 0.06 | 4.71[b] ± 0.16 | 1.97[c] ± 0.10 | 1 | *FAO/WHO (2011)* |
| TI (TiU/mg) | 8.43[a] ± 0.06 | 3.20[b] ±0.14 | 0.97[c] ± 0.17 | | |
| Alkaloids (mg/100 g) | 4.54[a] ± 0.06 | 1.80[b]± 0.014 | 0.97[c] ± 0.06 | | |
| Phenols (mgGAE/g) | 2.77[a] ± 0.06 | 1.10 ± 0.02 | 0.50[c] ± 0.02 | | |
| Flavonoids (mgQE/100 g) | 60.00[a] ± 3.53 | 40.00[b]± 0.53 | 52.50[a] ± 0.00 | | |

Notes:
* Data of the current study.
** Published references.
Key: RRSM, raw rubber seed meal; BRSM, boiled rubber seed meal; FRSM, fermented rubber seed meal.
Means with same superscripts in the same row are not significantly different ($p > 0.05$).

presented in Table 1 are some well-known referenced/published safety limits, which will be used to discuss the results herein (Refer to discussion section), specific to phytate (<25 mg/100 g), oxalate (<10 mg/100 g), tannin (12 mg/100 g), and HCN (1 mg/100 g) (*Abdoulaye, Brouk & Jie, 2011*; *FAO/WHO, 2011*; *Mark, 2020*; *Ndie & Okaka, 2018*). In general, the anti-nutrients/phytochemicals of processed (BRSM and FRSM) were significantly lower ($p < 0.05$) compared to the raw/unprocessed (RRSM) rubber seed meal samples. Overall, based on anti-nutrients/phytochemicals data, the rubber seed meals would trend: RRSM > BRSM > FRSM, except for the flavonoids, which would make it to trend: RRSM > FRSM > BRSM (Table 1).

## Variations in mineral contents

Variations in mineral contents of rubber seed meal samples are shown in Table 2. Across the RRSM, BRSM, and FRSM samples, the mineral contents (Mg, P, Ca, Fe, Zn, K, Na, Mn, Pb, and Se) differed significantly ($p < 0.05$). Specifically, FRSM obtained higher Mg (122.72 mg/100 g), P (228.93 mg/100 g), Ca (205.13 mg/100 g), Fe (7.86 mg/100 g), Zn (1.56 mg/100 g), K (800.16 mg/100 g), and Na (10.52 mg/100 g) compared to those of RRSM and BRSM. Peaks of Mn (0.41 mg/100 g), Pb (0.12 mg/100 g), and Se (0.03 mg/100 g) were found at RRSM. Additionally, the mineral contents of BRSM recorded the least values except for selenium (0.02 mg/100 g), which was above that of FRSM (0.00 mg/100 g). Across the RRSM, BRSM, and FRSM samples, the mineral composition obtained diverse ranges, as shown in Mg (from 112.15 to 122.72 mg/100 g), P (from 220.72 to 228.93 mg/100 g), Ca (from 193.57 to 205.13 mg/100 g), Fe (from 4.95 to 7.86 mg/100 g), Zn (from 0.99 to 1.56 mg/100 g), K (from 778.10 to 800.16 mg/100 g), Na (from 9.84 to 10.52 mg/100 g), Mn (from 0.35 to 0.41 mg/100 g), Pb (from 0,07 to 0.12 mg/100 g), and Se (from 0.00 to 0.03 mg/100 g), No Nickel (Ni) was detected. Also presented in Table 2 are some well-known referenced/published safety limits, which will be used to discuss the results herein (Refer to discussion section), specific to Mg (220–260 mg/100 g),

**Table 2 Variations in mineral contents of rubber seed meal samples.**

| Mineral element | *RRSM (mg/100 g) | *BRSM (mg/100 g) | *FRSM (mg/100 g) | **Safety limit (mg/100 g) | **Sources for the safety limits |
|---|---|---|---|---|---|
| Magnesium | 119.03[b] ± 0.88 | 112.15[c] ± 0.45 | 122.72[a] ± 1.50 | 220-260 | *FAO/WHO (2001)* |
| Phosphorus | 225.74[b] ± 0.60 | 220.72[c] ± 1.35 | 228.93[a] ± 0.54 | 700 | *FAO/WHO (2001)* |
| Calcium | 205.08[a] ± 3.72 | 193.57[b] ± 0.00 | 205.13[a] ± 0.33 | 1000 | *FAO/WHO (2001)* |
| Iron | 5.88[b] ± 0.23 | 4.95[c] ± 0.04 | 7.86[a] ± 0.23 | 8-18 | *FAO/WHO (2001)* |
| Zinc | 1.29[b] ± 0.06 | 0.99[c] ± 0.08 | 1.56[a] ± 0.00 | 8-11 | *WHO (2012a)* |
| Potassium | 795.28[a] ± 1.60 | 778.10[b] ± 5.05 | 800.16[a] ± 1.56 | 3500 | *WHO (2012a)* |
| Sodium | 10.21[b] ± 0.02 | 9.84[c] ±0.31 | 10.52[a] ± 0.04 | <2000 | *WHO (2012b)* |
| Manganese | 0.41[a] ± 0.01 | 0.35[b] ± 0.01 | 0.39[ab] ± 0.01 | 1.8-2.3 | *NIH (2020)* |
| Lead | 0.12[a] ± 0.00 | 0.07[c] ± 0.01 | 0.10[b] ± 0.01 | | |
| Selenium | 0.03[a] ± 0.00 | 0.02[a] ± 0.00 | 0.00[b] ± 0.00 | 0.055 | *Institute of Medicine (2000)* |
| Nickel | ND | ND | ND | | |

Notes:
* Data of the current study.
** Published references.
Means with same superscripts in the same row are not significantly different ($p > 0.05$).
Key: RRSM, raw rubber seed meal; BRSM, boiled rubber seed meal; FRSM, fermented rubber seed meal; ND, not detected

P (700 mg/100 g), Ca (1,000 mg/100 g), Fe (8–18 mg/100 g), Zn (8–11 mg/100 g), K (3,500 mg/100 g), Na (<2,000 mg/100 g), Mn (1.8–2.3 mg/100 g), and Se (55 μg) (*FAO/ WHO, 2001*; *Institute of Medicine, 2000*; *NIH, 2020*; *WHO, 2012a, 2012b*). Overall, based on quantities, the trend of mineral content across samples would follow: FSRM > RRSM > BRSM, except for selenium, which would make it to trend: RRSM > BRSM > FRSM (Table 2). The rubber seed meal can be considered as fortified with mineral elements (Mg, P, Ca, Fe, Zn, K, Na, Mn, Pb, and Se).

## Variations in vitamin contents

Variations in vitamin contents of rubber seed meal samples are shown in Table 3. Across the RRSM, BRSM, and FRSM samples, the vitamin contents (vitamin B1, B2, B3, C, E, and β-carotene) differed significantly ($p < 0.05$). Additionally, the vitamin contents obtained diverse ranges across the RRSM, BRSM, and FRSM samples, as shown in vitamin B1 (from 0.21 to 0.30 mg/100 g), vitamin B2 (from 0.31 to 0.52 mg/100 g), vitamin B3 (from 0.51 to 0.77 mg/100 g), vitamin C (from 10.03 to 29.09 mg/100 g), vitamin E (from 5.92 to 8.90 mg/100 g), and β-carotene (from 2.09 to 3.77 mg/100 g). Considering the vitamin content data, those of RRSM obtained the highest values whereas those of BRSM obtained the least values. Additionally, the vitamin content values in FRSM were significantly higher ($p < 0.05$) compared to those of BRSM. Overall, based on quantities, the trend of vitamin content across samples would follow: RRSM > FSRM > BRSM. Also presented in Table 3 are some well-known referenced/published safety limits, which will be used to discuss the results herein (Refer to discussion section), specific to vitamin B1 (1.1–1.2 mg/100 g), B2 (1.1–1.3 mg/100 g), B3 (14–16 mg/100 g), C (60 mg/100 g), E (15 mg/100 g), and β-carotene (0.7–0.9 mg/100 g) (*Finglas, 2000*; *Kirkland & Meyer-Ficca, 2016*; *Kubala, 2018*; *Institute of Medicine, 2000*; *NIH, 2020*).

**Table 3 Variations in vitamin composition (mg/100g) of rubber seed meal sample.**

| Vitamin content | *RRSM (mg/100 g) | *BRSM (mg/100 g) | *FRSM (mg/100 g) | **RDA (Adults) (mg/100 g) | **Sources for RDA (Adults) |
|---|---|---|---|---|---|
| B1 (Thiamin) | $0.30^a \pm 0.01$ | $0.21^b \pm 0.01$ | $0.26^a \pm 0.01$ | 1.1-1.2 | *Finglas (2000)* |
| B2 (Riboflavin) | $0.52^a \pm 0.01$ | $0.31^c \pm 0.03$ | $0.38^b \pm 0.00$ | 1.1-1.3 | *Finglas (2000)* |
| B3 (Niacin) | $0.77^a \pm 0.05$ | $0.51^b \pm 0.02$ | $0.72^a \pm 0.04$ | 14-16 | *Kirkland & Meyer-Ficca (2016)* |
| β-carotene | $3.77^a \pm 0.02$ | $2.09^c \pm 0.09$ | $3.22^b \pm 0.09$ | 0.7-0.9 | *Kubala (2018)* |
| Vitamin C | $29.09^a \pm 0.52$ | $10.03^c \pm 0.60$ | $15.09^b \pm 0.43$ | 60 | *Institute of Medicine (2000)* |
| Vitamin E | $8.90^a \pm 0.32$ | $5.92^c \pm 0.23$ | $7.46^b \pm 0.04$ | 15 | *NIH (2020)* |

Notes:
* Data of the current study.
** Published references.
Means with same superscripts in the same row are not significantly different ($p > 0.05$).
Key: RRSM, raw rubber seed meal; BRSM, boiled rubber seed meal; FRSM, fermented rubber seed meal.

# DISCUSSION

The phytate range of rubber seed meal (from 6.97 to 19.62 mg/100 g) (Table 1), was below those *Popova & Mihaylova (2019)* reported for legume grains (386 to 714 mg/100 g), nearer to those *Abdulhamid, Ibrahim & Warra (2014)* reported for *Egusi* (3.09 mg/100 g), and falls below <25 mg/100 g safety threshold applied to edible grains (*Abdoulaye, Brouk & Jie, 2011*). As an anti-nutritive agent, phytic acid blocks absorption and limits the bioavailability of such minerals as Fe, Zn, and Ca (*Urbano et al., 2000*). Oxalates, on the other hand, bind to minerals like Ca, Fe, and Mg to form insoluble compounds, which brings some concerns given that calcium oxalate has been implicated in kidney stones, and render it unavailable for absorption (*Spritzler, 2017*). The oxalate range of rubber seed meal (from 3.36 to 13.26 mg/100 g) (Refer to Table 1) fell below those *Mark (2020)* reported for (boiled) soybeans (52 mg/100 g) but compares well with those of Mung beans (8 mg/100 g). *Mark (2020)* considered foods containing >10 mg oxalate as high-oxalate food, consistent with the classification of the American Dietetic Association(ADA). Based on this argument, we opine that the processed rubber seed meal herein is low in oxalates.

The tannin range of rubber seed meal (from 0.92 to 8.98 mg/100 g) (Refer to Table 1) was below those *Popova & Mihaylova (2019)* reported for legumes (soybeans, peanuts, and beans) (1.8–18 mg/100 g) and compares well with *Egusi* (6.19 mg/100 g) (*Abdulhamid, Ibrahim & Warra, 2014*). The stipulated maximum limit of tannin in food is 12 mg/100 g (*Ndie & Okaka, 2018*). Above this (maximum limit of tannin), the protein absorption can be compromised, and if this were to happen in a consumer that ingested food containing processed rubber seed meal, which comprised such quantities of tannin, the intestinal wall could be damaged. Based on this argument, the tannin of rubber seed meal herein can be considered below the tannin safety limits. The saponin range of rubber seed meal (from 1.60 to 4.91 mg/100 g) (Refer to Table 1) was below those *Banaskiewicz (2011)* reported for soybean meal (60 mg/100 g) and those *Russo & Reggiani (2016)* reported for linseed meal (17 mg/100 g). Saponins, despite their haemolytic activity and bitter taste which limits their palatability, still show cholesterol-lowering and anticancer properties as health benefits and remain commercially significant given their applicability across food and pharmaceutical industries (*GucluUstundag & Mazza, 2007*).

The HCN range of rubber seed meal (from 1.97 to 12.41 mg/100 g) (Refer to Table 1) fell below those *Obiakor-Okeke (2014)* reported for raw lima beans flour (38.38–43.55 mg/100 g), those *Khan et al. (2010)* reported for linseed (31.05 mg/100 g), and those *Farr et al. (2019)* reported for roasted (14.30 mg/100 g) rubber seed meal. Results herein showed that FRSM recorded the highest percentage reduction in HCN (84.13%) while BRSM had the least (62.05%) percentage reduction in HCN. The reduced HCN in rubber seed could be due to inactivation of glycosidase, leaching, and vaporization of HCN after it is formed from cyanogenic glycosides hydrolysis and/or the destruction of the cyanogenic glycoside compounds (*Feng, Shen & Chavez, 2003*). Probably, the combination of boiling and fermentation in FRSM helped to reduce the HCN, which agrees with the report of *Daulay, Adelina & Suharman (2014)*. Differences in nature (flour, liquid, solids, gruel, coarse, etc.) of associated foods can also result in HCN variations. Moreover, the safety limit of cyanide differs across foods. For instance, maximum levels of HCN in cassava flour (1 mg/100 g), garri (0.2 mg/100 g), nougat and marzipan (5 mg/100 g), alcoholic beverages (3.5 mg/100 g), and canned stone fruits (0.5 mg/100 g) have already been established by the Codex Alimentarius Commission (*FAO/WHO, 2011*; *EFSA, 2016*; *Schrenk et al., 2019*). Interestingly, there appears to be no regulation that has stipulated the cyanide levels in foods (*EFSA, 2016*). HCN variations found in the processed rubber seed meals herein might suggest that the processing methods did not sufficiently detoxify/eliminate the cyanogenic contents. *Sharma, Saha & Saha (2014)* equally emphasised that there is a great need to develop economical ways of detoxifying cyanogenic-containing food products, especially removing up to 85% of the toxicant (HCN). However, considering that the lethal dose of cyanide is 0.5 to 3.5 mg/kg body weight (bw) (*EFSA, 2016*), a 70 kg of weight adult, for example, would have to consume about 35 mg of HCN to become affected, which would be based on the minimum lethal dose of cyanide of 0.5 mg/kg bw, and such would equal to 1776 g of FRSM and 743 g of BRSM, and this is most unlikely to happen for any daily consumption. In the view to reduce the HCN content of FRSM below toxic level, increased hours of boiling and fermentation of the rubber seed should strongly be recommended for it to be used as food and feed.

The TI activity range of processed rubber seed meal (0.97 to 8.43 TiU/mg) (Refer to Table 1) fell below those *Yalcin & Basman (2015)* reported for soybeans (94.1 TiU/mg) but competes well with those *Embaby (2010)* reported for peanut (5.60 TiU/mg). The TI activity decreased significantly ($p < 0.05$) in the processed rubber seed meal (8.43 to 0.97 TiU/mg) (Table 1), recording a percentage reduction of 62.04% for BRSM and 88.50% for FRSM. This appears not consistent with the report of *Udo, Ekpo & Ahamefule (2018)* and *Farr et al. (2019)*, who demonstrated heat processing to deliver up to 100% reduction in TI activity. Variations in TI activity of the rubber seed meal herein probably arose due to varietal differences (of product), harvest period, and differences in processing methods/treatments. Besides, we could not find any recorded safety limits of TI activity in foods. Potentially, such conventional processing like fermentation and heat treatments could deactivate the TI activity.

Alkaloids are important chemical compounds, with strong biological effects on humans, even at very small doses (*Kurek, 2019*). Herein, the alkaloid content range of processed

rubber seed meal (0.97 to 4.54 mg/100 g) (Table 1) compares favourably with those *Okwu & Orji (2007)* reported for soy and cowpea seeds (1.49–1.64 mg/100 g). Known for their bitter taste, alkaloids could perform some anti-inflammatory and analgesic functions, and serve as a local anaesthetic for pain relief, supplemented by their antimicrobial and antifungal properties (*Kurek, 2019*). On the other hand, the total phenolic content range of processed rubber seed meals (0.50–2.77 mg GAE/g) (Refer to Table 1) compares favourably with that *Mujic et al. (2011)* reported for soybean (0.87 to 2.16 mg GAE/g). Phenols are important in enhancing the nutritional value in foods, acting as antioxidants (that can prevent cellular damage due to free radical oxidation reactions) as well as anti-inflammatory agents (*Reis-Giada, 2013*; *Ofoedu et al., 2021*). The flavonoids content of RRSM (60.00 mg QE/100 g), BRSM (40.00 mg QE/100 g), and FRSM (52.50 mg QE/100 g) (Table 1) compares favourably with those of soybean (51 mg/100 g) (*Josipovic et al., 2016*). Unlike the trend reported in the other phytochemicals of the current study, the FRSM obtained about a 30 % increase in flavonoids over the BRSM (Refer to Table 1), and given such (increases in flavonoids), the fermentation process might improve the antioxidant activity (*Hur et al., 2014*). A significant increase in flavonoids with okra fermentation has been reported elsewhere (*Adetuyi & Ibrahim, 2014*). The undesirable bitter taste of flavonoids may limit its use as food bioactive, despite being associated with a broad spectrum of health-promoting effects, for example, anti-oxidative, anti-inflammatory, anti-mutagenic, and anti-carcinogenic effects (*Panche, Diwan & Chandra, 2016*).

Indeed, the anti-nutrient composition of food products could be impacted by processing methods (*Udo et al., 2018*). Comparably, the reduction in anti-nutrient levels of grains/ legumes could arise from conventional processing methods, like soaking, dehulling, boiling, pressure cooking, germination, and fermentation (*Patterson, Curran & Der, 2016*). Herein, the boiling might have significantly ($p < 0.05$) reduced the anti-nutrient/ phytochemical contents in BRSM, specifically between 43–75%. In like manner, the combined action of boiling and fermentation process might have significantly ($p < 0.05$) reduced the anti-nutrient/phytochemicals in FRSM, specifically between 64–89%. The combined effects of boiling and fermentation, in varying degrees, might be contributing to some of the reductions in anti-nutrients/phytochemicals in these rubber seed meals, which could happen as the enzyme gets released by microbial-led fermentation (*Sokrab, Ahmed & Babiker, 2014*). Boiling, as a processing method, would, not only hydrate the product matrix but also, induce the leaching out of water-soluble anti-nutrients/phytochemicals (*Nwosu et al., 2013*).

Feasibly, the processing methods might have influenced the mineral losses/retention in the processed rubber seed meals, either through leaching process during boiling, from the samples into the processing water (*Gokoglu, Yerlikaya & Cengiz, 2004*; *Ihediohanma et al., 2014*; *Osuji, Ofoedu & Ojukwu, 2019*), or (solid-state) fermentation, which markedly improves the nutritive values of food (*Ibeabuchi et al., 2014*), or reductions in the overall anti-nutrients that determines the (actual) mineral contents of seed meal samples (*Kumar et al., 2010*). The Ca range (193.57–205.13 mg/100 g) obtained herein (Refer to Table 2) fell below the 1,000 mg/day recommended dietary allowance (RDA) of *FAO/ WHO (2001)*. Besides, Ca occurring in (rubber seed meal) products might be facilitating

both absorption and bioavailability of minerals like Mg and Zn (*Agbaje et al., 2014*; *Ofoedu et al., 2020*). Additionally, muscle function, vascular contraction/vasodilation, nerve transmission, and hormonal secretions within the human body are associated with Ca (*Institute of Medicine, 2010*).

As a cofactor of many enzyme systems, Mg can regulate diverse biochemical reactions as well as physiological activities (*Agbaje et al., 2014*; *Rude, 2014*). Herein, the range of Mg of processed rubber seed meal (112.15–122.72 mg/100 g) (Refer to Table 2) fell below the WHO dietary recommended daily intake (RDI) (*FAO/WHO, 2001*), which makes it a fair source (of Mg). In like manner, the range of Zn in the processed rubber seed meals (0.99 to 1.56 mg/100 g) (Table 2) fell below the RDI for males (11 mg/day) and females (8 mg/day), which is required for DNA synthesis/cell division, enzyme-catalytic activities, immune function, protein synthesis, and wound healing (*Institute of Medicine, 2001*). The P range in processed rubber seed meals (220.72 to 228.93 mg/100 g) (Refer to Table 2), seemed about one-third of the WHO dietary RDI of 700 mg/day (*WHO, 2012a*). The P remains among the human body's essential energy source (ATP) and is very needed in bones, teeth, and cells (*Heaney, 2012*).

As an essential part of haemoglobin and myoglobin, Fe enhances physical growth, neurological development, cellular function, and hormonal synthesis, and facilitates healthy connective tissue and muscle metabolism (*Wessling-Resnick, 2014*). Herein, the Fe range in the processed rubber seed meals (4.95 to 7.86 mg/100 g) (Table 2) compares favourably with RDI for men (8 mg/day) and women (18 mg/day) (*WHO, 2012b*). Additionally, the K range in the processed rubber seed meal (778.10 to 800.16 mg/100 g) (Refer to Table 2), although compared favourably with 813.10 mg/100 g of linseed (*Sayer, 2018*), but appears lower than the RDI recommended for adults (3,500 mg/day) (*WHO, 2012b*). In the human body, K plays a significant role in muscle contractions, heart function, and managing water balance (*Pohl, Wheeler & Murray, 2013*). Herein, the Na range in the processed rubber seed meal (9.84 to 10.52 mg/100 g) (Table 2) appears lower than those *VanEys, Offner & Bach (2004)* reported for soybeans (29 mg/100 g) and *Sayer (2018)* reported for linseed (30 mg/100 g). Reduced Na intake of less than 2,000 mg/day as recommended by WHO, would reduce the risks of high blood pressure, cardiovascular disease, stroke, and coronary heart diseases (*WHO, 2012b*). The Na in the processed rubber seed meal, herein, is quite low, compared to the WHO recommended limit of 2,000 mg/day (*WHO, 2012b*).

More so, Se is an important trace mineral essential for the proper functioning of the human body, given its significant role in metabolism and thyroid function, to protect the body from damage caused by oxidative stress (*Kubala, 2019*). Herein, the Se range (0.02 to 0.03 mg/100 g) (Refer to Table 2) in rubber seed meal resembled values of other oil seeds such as peanut (0.0313 mg/100 g), corn (0.0332 mg/100 g), sunflower (0.0225 mg/100 g), and soybean (0.0458 mg/100 g) (*Dugu et al., 2003*). As a trace mineral required by the body in small amount, the amount of Se in the rubber seed meal compared favourably with the RDA of 0.0055 to 0.0070 mg per day (that is, 0.0055 mg/day for adult men and women, 0.0065 mg/day for lactating women and 0.0070 mg/day for pregnant women (*Winther et al., 2017*; *Anonymous, 2021*; *Institute of Medicine, 2000*). Processing

methods, however, reduced the amount of Se in the rubber seed meal, as the BRSM recorded 0.02 mg/100 g, whereas it (the Se) might have been lost in the FRSM (0.00 mg/ 100 g). Utilising boiling method to process the rubber seeds, despite the latter's specific antioxidant property, appears to be efficient in retaining some of the Se in the rubber seed meal.

Vitamins (B1, B2, B3, C, E, and β-carotene) across the RRSM, BRSM, and FRSM samples (Refer to Table 3) resemble those reported in soybean, sunflower, rapeseed, cotton, and groundnut (*Muhammad et al., 2013*). Ranges of vitamins B1 (0.21 to 0.30 mg/ 100 g), B2 (0.31 to 0.52 mg/100 g), and B3 (0.51 to 0.77 mg/100 g) across the rubber seed meals, (Table 3) fell below the RDA standards of vitamin B1, B2, and B3 for male (1.2 mg/ day, 1.3 mg/day, and 16 mg/day) and female (1.1 mg/day, 1.1 mg/day, and 14 mg/day) respectively (*Cronkleton, 2019*), which makes them a fair source of the B vitamins. Typically, the B vitamins help the body's cells to process the nutrients (carbohydrate, protein, and fat) into energy, as well as ensure normal nervous/physiological functions (*Sachdev & Shah, 2020*; *Bradford, 2015*). The β-carotene is a precursor of vitamin A (retinol) since the human body converts β-carotene to retinol (*Newman, 2017*). The rubber seed meals had β-carotene (Provitamin A) values of 3.77 mg/100 g for RRSM, 2.09 mg/ 100 g for BRSM, and 3.22 mg/100 g for FRSM (Refer to Table 3). Therefore, to convert the β-carotene into its retinol equivalent requires that 0.006 mg β-carotene would equal 0.001 mg retinol (*Scott & Rodriquez, 2000*; *EFSA, 2015*). Therefore, the retinol of the RRSM (3.77 mg/100 g), BRSM (2.09 mg/100 g), and FRSM (3.22 mg/100 g) would equal 0.6283 mg/100 g, 0.3483 mg/100 g, and 0.5367 mg/100 g, respectively. Based on the recommended dietary allowance (RDA) of vitamin A for adult men (0.9 mg) and women (0.7 mg) (*Kubala, 2018*), the rubber seed meal herein suggests to be a good source of provitamin A. Additionally, β-carotene remains an antioxidant that protects the body from free radicals (*Newman, 2017*).

Given the range of vitamin C (10.03 to 29.09 mg/100 g) and vitamin E (5.92 to 8.90 mg/ 100 g), the BRSM obtained the least (vitamins C and E) values (Refer to Table 3). The daily recommended dietary allowance is 60 mg for vitamin C (*Institute of Medicine, 2000*) and 15mg for vitamin E (*NIH, 2020*). Thus, the rubber seed meal is a fair source of vitamin E, but a poor source of vitamin C. Nonetheless, processing (boiling and fermentation) methods might have affected the vitamins C and E herein. Vitamin losses have been associated with the leaching of soluble components (*Osuji, Ofoedu & Ojukwu, 2019*; *Ihediohanma et al., 2014*) into the processing water. Though (rubber seed meal) products from FRSM were subjected to boiling prior to fermentation, *Eyarkai, Chandrasekar & Karthikeyan (2017)* reported that (solid-state) fermentation could improve the nutritive value (vitamins) of the food product. The significantly higher ($p < 0.05$) vitamin C and E in FRSM, compared to BRSM (Table 3), could be that microbial action potentially enhanced these specific vitamins (C and E).

## CONCLUSIONS

The changes in anti-nutrient, phytochemical, and micronutrient contents of different processed rubber seed meals were successfully investigated. For emphasis, the rubber seeds

underwent processing, which brought about three seed meal flour groups, i.e., raw, boiled, and fermented rubber seed meals. Processing methods were shown to markedly influence the studied anti-nutrient/phytochemical and micronutrient (vitamins and minerals) contents of the rubber seed meals. Specifically, the combined action of boiling and fermentation processes effectively brought about some noticeable reduction in anti-nutrients. Despite that FRSM obtained significantly higher mineral content compared to the other groups (RRSM and BRSM), it also obtained the least phytochemical/anti-nutrient composition, except for flavonoid. The FRSM obtained a higher vitamin content, after RRSM. Importantly, the processing methods herein have not yet succeeded in removing HCN in the (processed) rubber seed meals, but can be seen to be heading toward the right direction.

Given that the rubber seed is a source of micronutrients in an appreciable amount, fermentation seems to have enriched the nutritive value of its seed meal. The combined action of boiling and fermentation should be recommended for the proper utilisation of rubber seed as food/feed and in the production of edible oil. It should also be recommended that processed rubber seed meals be used in proportionate amounts during food product formulation to enhance the nutrient content of the product as well as mask the effect of such anti-nutrients like HCN and TI. Further, the rubber seeds should be exposed to increased (high) temperature treatment, such as prolonged boiling before fermentation, which can help enhance the detoxification of toxicants, that may be carried over into the processed seed meal. Subsequently, this research lays the foundation for future studies towards the determination of the toxicity levels, and nutrient retention of rubber seeds processed by either boiling and fermentation, or non-thermal processing techniques, like ultrasound, cold plasma, pulsed electric field, high-pressure processing, and pulsed light.

## ABBREVIATION

| | |
|---|---|
| **RRSM** | Raw rubber seed meal |
| **BRSM** | Boiled rubber seed meal |
| **FRSM** | Fermented rubber seed meal |
| **Mg** | Magnesium |
| **P** | Phosphorus |
| **Ca** | Calcium |
| **Fe** | Iron |
| **Zn** | Zinc |
| **K** | Potassium |
| **Na** | Sodium |
| **Mn** | Manganese |
| **Pb** | Lead |
| **Se** | Selenium |

| Vit B1 | Thiamine |
|---|---|
| **Vit. B2** | Riboflavin |
| **Vit B3** | Niacin |
| **Vit C** | Ascorbic acid |
| **Vit E** | Tocopherol |
| **HCN** | Hydrogen cyanide |
| **TI** | Trypsin inhibitor |
| **RDA** | Recommended dietary allowance |
| **RDI** | Recommended daily intake |

### Funding
Publication financed by the project UPWR 2.0: international and interdisciplinary programme of development of Wrocław University of Environmental and Life Sciences, co-financed by the European Social Fund under the Operational Program Knowledge Education Development, under contract No. POWR.03.05.00-00-Z062/18 of June 4, 2019. The funders had no role in study design, data collection and analysis, decision to publish, or preparation of the manuscript.

### Grant Disclosures
The following grant information was disclosed by the authors:
Wrocław University of Environmental and Life Sciences.
Operational Program Knowledge Education Development: POWR.03.05.00-00-Z062/18.

### Competing Interests
Charles Odilichukwu R. Okpala is an Academic Editor of PeerJ.

### Author Contributions
- Chidinma M. Agbai conceived and designed the experiments, analyzed the data, prepared figures and/or tables, and approved the final draft.
- Ijeoma A. Olawuni conceived and designed the experiments, prepared figures and/or tables, and approved the final draft.
- Chigozie E. Ofoedu conceived and designed the experiments, analyzed the data, authored or reviewed drafts of the paper, and approved the final draft.
- Chidi J. Ibeabuchi performed the experiments, prepared figures and/or tables, and approved the final draft.
- Charles Odilichukwu R Okpala analyzed the data authored or reviewed drafts of the paper, and approved the final draft.
- Ivan Shorstkii analyzed the data, authored or reviewed drafts of the paper, and approved the final draft.
- Małgorzata Korzeniowska analyzed the data, authored or reviewed drafts of the paper, and approved the final draft.

## Data Availability

The raw data contains the duplicate measurements and are available in the Supplementary File.

## Supplemental Information

Supplemental information for this article can be found online at http://dx.doi.org/10.7717/peerj.11327#supplemental-information.

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
