# Peer review of "Changes in anti-nutrient, phytochemical, and micronutrient contents of different processed rubber (Hevea brasiliensis) seed meals"

_PeerJ, doi:10.7717/peerj.11327_

## Round 0.1 · original submission · Major Revisions

Please take into consideration the reviewer’s comments and provide back a point-by-point rebuttal letter addressing those concerns.

·

Basic reporting

The manuscript presents valuable findings of composition changes in processed rubber seed meals. The general structure can be easy to follow, but upon careful reading, there are some issues that I would like to comment.

Line 28-29 (Abstract): "...rubber seeds are largely underutilized given its perception as inedible/toxic, therefore the quest for product diversification of rubber seed."
I don’t understand the logic of this sentence. Maybe “product diversification” is not the right term.

The Abstract is generally self-contained (except L.28-29), however, I am wary about conclusion relating hydrogen cyanide (see my comment on Validity of the Findings).

The Introduction provides an overview of the chemical composition of rubber seed. However, the discussion of rubber seed processing is lacking. The writing also needs improvement:
- Several literatures cited in the Introduction (e.g. Akinmutimi, 2004; Nwokolo, 1996; Lukman et al, 2018; Siti et al, 2013) are not included in the References
- Line 56-58: "Importantly, that excess of 25% (...) protein-rich foods." Citation needed.
- For information that may constantly change, (Nwokolo, 1996) seems like an outdated reference for rubber plantation area and rubber seed production. Maybe it provides rubber seed yield (tonnes seed/ha), but since it is not on the reference list, I cannot evaluate for myself. In any case, the data for the plantation area should be referenced with recent literature.
- Line 71-77: This part lacks coherence. In the beginning, the statement is vague ("high essential nutritive value", "considerable amounts"), but the next sentence contains very precise numbers (two decimal digits). The author also mentions rubber seed meal, rubber seed protein isolates, and rubber seed oil. It must be clear what the authors mean by these, and how they are connected to the previous sentence. I would suggest the author to re-write this part, substantiate the vague statements with numbers and/or compare with "conventional seed meals".
- Line 91-93: "...the processing of rubber seed can involve a range of activities, such as, boiling, fermentation, roasting, as well as other forms of heat treatment." This should be more elaborate. As I mentioned before, there should be more critical overview in the Introduction that leads to the reason for selecting processing methods.

Results and Discussion should be presented in reasonable order (following order in tables, or otherwise any reasonable groupings). If the authors choose a certain grouping in the discussion (e.g., components that increase, components that decrease), it will help the reader if the paragraph starts by explaining what will be discussed, e.g "The contents of component X, Y, Z increased after processing"

In the discussion, whether a value is a requirement (minimum) or limit (maximum) should be clear to evaluate quality. Specific comments:
- Line 332-334: from the sentence, I think it should be tannin maximum limit instead of minimum
- Line 333, 349: I don't think it is necessary to state the unit twice: 120 mg/kg (equivalent to 12 mg/100g). Just use mg/100 g so it is consistent throughout Discussion.
- Serving portion should be indicated for evaluation of RDI/RDA.

I would suggest that the discussion on anti-nutrient and phytochemicals is divided into separate sections. This is related to my previous comment, because the evaluation of anti-nutrient should refer to the maximum limit. On the other hand, discussion on phytochemicals is not as strict because there are potential benefits in some compounds.

Line 472: Please clarify, is RDA for B vitamins for the sum of B1, B2, B3 or for each?

RDI and RDA should be added in the Abbreviation list.

Figure 1 is too general. The explanation in Line 110-114 and Figure 2 is adequate. I would suggest removing Figure 1.

Figure 2 needs revision:
- Raw rubber seed (meal is missing)
- Fermentation (3--days in other line)
- I think there should be dry milling step after oven drying and fermentation

Tables can be more informative (self-contained): I would suggest adding one column in each table as reference values (maximum/safe limit for anti-nutrient, RDI/RDA for phytochemical and micronutrients).

Table 1: Phenols in BRSM has no superscript

Table 2: Why is selenium not discussed? What is the meaning of dash (-) for nickel? If not detected, please state.

Experimental design

No comment

Validity of the findings

The result is promising as a basis for practical application. However, I have some comments as follows:

The Methods mention that the results are from triplicate determination, however, the submitted data only show duplicate results. Perhaps the author can correct this discrepancy.

I think there should be a comparison with other processing methods of rubber seed. The authors already cite at least two references (Aguihe et al, 2017; Udo et al, 2018) that discuss similar RSM components. However, there is only one mention of Udo et al in the discussion about TI. There might be other relevant literature that has not been referred to. The comparison would also help explain how the process influences the composition.

I find the authors' conclusion on hydrogen cyanide is problematic. First of all the authors already mentioned that the safety limit of HCN is 1 mg/100 g, which could not be met by any of the processes although FRSM came close. (note: I also think L348 should be maximum limit, not minimum). Second of all, the authors conclude that the processed RSM are safe based on lethal dose. I believe the unpleasantness caused by HCN would occur way below the lethal dose. I would suggest the authors presenting a more convincing argument for the safety of processed RSM. Otherwise, I happily accept that the current process has not yet succeeded in removing HCN, but already heading toward the right direction.

Line 520: "Given that the rubber seed is a poor source of micronutrients." I find this statement comes out of nowhere. The way the authors discussing the result, it is difficult to reach this conclusion.

Additional comments

Just a suggestion, I think it would help if there is a graph (e.g radar charts) that compare the three processed RSM together without overly repeating the results.

Reviewer 2 ·

Basic reporting

In general, the authors had done a good job in writing the article.

Experimental design

Good experimental design

Validity of the findings

No comments; good findings

Additional comments

The manuscript is written well except for minor grammatical errors in the text. Overall, this paper can be accepted.

Annotated reviews are not available for download in order to protect the identity of reviewers who chose to remain anonymous.

---

## Round 0.2 · Minor Revisions

Please be encouraged by the prompt review, and we look forward to your revised manuscript. Aside from the reviewer comments, please check the resolution of figures 1 and 2, since they appear slightly blurred in the proofs PDF. Also, is highly recommended that you use a professional English proofreading service to edit style since there are issues to be addressed.

·

Basic reporting

The authors have made an excellent revision and addressed almost all of major issues in the previous version. The English writing still needs some work to improve clarity.

Aside of the English, I only have some minor comments:

- L25-27 (Abstract): Maybe add "...to produce edible oil and protein" since the next sentence mention "protein source"

- L321: This sentence still mention "triplicate measurement"

- L445-446: I think it should be "...equal to 1776 g of FRSM and 743 g of BRSM"

- Table 2: safety limit units need to be clarified (whether mg/100 mg on the heading or mg and µg)

- There seems to be differences of alkaloid values for FRSM between Table 1 and supplementary file. This might be due to glitch in Excel, but please check.

Experimental design

No comment

Validity of the findings

No comment

---

## Round 0.3 · accepted · Accept

Thanks for addressing the minor revisions requested. Now your manuscript is accepted in PeerJ.